# Improved Long-Term Preservation of *Cannabis* Inflorescence by Utilizing Integrated Pre-Harvest Hexanoic Acid Treatment and Optimal Post-Harvest Storage Conditions

**DOI:** 10.3390/plants13070992

**Published:** 2024-03-30

**Authors:** Matan Birenboim, Daniel Chalupowicz, David Kenigsbuch, Jakob A. Shimshoni

**Affiliations:** 1Department of Food Science, Institute for Postharvest and Food Sciences, Agricultural Research Organization, Volcani Center, Rishon LeZion 7505101, Israel; 2Department of Plant Science, The Robert H. Smith Faculty of Agriculture, Food and Environment, The Hebrew University, Rehovot 7610001, Israel; 3Department of Postharvest Science, Institute for Postharvest and Food Sciences, Agricultural Research Organization, Volcani Center, Rishon LeZion 7505101, Israel

**Keywords:** *Cannabis sativa* L., hexanoic acid, vacuum storage, cannabinoids, terpenes

## Abstract

The effort to maintain cannabinoid and terpene levels in harvested medicinal cannabis inflorescence is crucial, as many studies demonstrated a significant concentration decrease in these compounds during the drying, curing, and storage steps. These stages are critical for the preparation and preservation of medicinal cannabis for end-use, and any decline in cannabinoid and terpene content could potentially reduce the therapeutic efficacy of the product. Consequently, in the present study, we determined the efficacy of pre-harvest hexanoic acid treatment alongside four months of post-harvest vacuum storage in prolonging the shelf life of high THCA cannabis inflorescence. Our findings indicate that hexanoic acid treatment led to elevated concentrations of certain cannabinoids and terpenes on the day of harvest and subsequent to the drying and curing processes. Furthermore, the combination of hexanoic acid treatment and vacuum storage yielded the longest shelf life and the highest cannabinoid and mono-terpene content as compared to all other groups studied. Specifically, the major cannabinoid’s—(-)-Δ9-trans-tetrahydrocannabinolic acid (THCA)—concentration decreased by 4–23% during the four months of storage with the lowest reduction observed following hexanoic acid pre-harvest treatment and post-harvest vacuum storage. Hexanoic acid spray application displayed a more pronounced impact on mono-terpene preservation than storage under vacuum without hexanoic acid treatment. Conversely, sesqui-terpenes were observed to be less prone to degradation than mono-terpenes over an extended storage duration. In summation, appropriate pre-harvest treatment coupled with optimized storage conditions can significantly extend the shelf life of cannabis inflorescence and preserve high active compound concentration over an extended time period.

## 1. Introduction

*Cannabis sativa* L., the sole species in the *Cannabaceae* family, is an annual herb with proven therapeutic benefits for ameliorating medical conditions such as pain, epilepsy, and cancer [1,2,3,4]. Despite various available medicinal cannabis products, dried inflorescence remains among the most popular primary medical products. The plant’s therapeutic effects are attributed to its secondary metabolites: over 120 cannabinoids and terpenes [5,6]. These compounds, particularly the volatile terpenes, contribute to the “entourage effect”, making their preservation over a long period of time crucial for both medicinal and recreational users [7]. Given the chemical variability between different cannabis genotypes, the term “chemovar”, encompassing the full cannabinoid and terpene profile, is preferred for classification [8,9,10].

During the flowering stage, the first cannabinoid synthesized in the cannabinoid biosynthetic pathway is cannabigerolic acid (CBGA) [11,12]. CBGA is synthesized from geranyl pyrophosphate (GPP) and olivetolic acid (OA) by the enzyme CBGA-synthase [6,11,12]. CBGA is the precursor for three main acidic cannabinoids, namely, (-)-Δ9-trans-tetrahydrocannabinolic acid (THCA), cannabidiolic acid (CBDA), and cannabichromenic acid (CBCA), which are synthesized by the three different enzymes, THCA-synthase, CBDA-synthase, and CBCA-synthase, respectively [6,11,12]. Hexanoic acid is a derivative of hexanoyl CoA, one of the key precursors involved in the formation of OA in the cannabis plant [11,12]. Multiple research studies have elucidated the efficacy of hexanoic acid in enhancing plant resistance against pathogenic organisms, notably *Botrytis cinerea* and *Alternaria alternata* [13,14,15]. This effect has been observed in a range of plant species, including tomato, arabidopsis, and various citrus species [13,14,15]. A significant outcome of this enhanced resistance is the observed reduction in the diameter of lesions caused by these pathogens [13,14,15]. Furthermore, recent investigations have shed light on the underlying mechanism of hexanoic acid’s priming activity in citrus plants [13]. It has been demonstrated that hexanoic acid modulates the biosynthetic pathways of mevalonic acid and linolenic acid. This modulation leads to an augmented production of volatile compounds in the treated plants, indicating a complex interaction with the plant’s metabolic processes [13]. The priming effect of hexanoic acid was also reported by Vicedo et al., demonstrating that the treatment of tomatoes with hexanoic acid led to an elevated concentration of the secondary metabolite caffeic acid [15]. Hence, we propose that the application of hexanoic acid as a pre-harvest priming agent on cannabis inflorescences could potentially lead to an augmentation in cannabinoid concentrations.

After the initial stages of post-harvest treatments which include trimming and a 2–3 weeks of drying and curing, the dried cannabis inflorescences are ready for dispensing [1,9,16]. Proper storage conditions in designated packages greatly affect the cannabis inflorescences’ shelf life, which is defined as the time during which the initial concentrations of the major cannabinoids THCA (in THCA chemovars) and CBDA (in CBDA chemovars) are ≥95% of their initial levels [17]. Based on the existing literature, it is understood that both THCA and CBDA undergo various degradation pathways during storage [6]. Specifically, THCA may degrade into cannabinolic acid (CBNA) or cannabitriolic acid (CBTA) through oxidation, transform into (-)-Δ8-trans-tetrahydrocannabinolic acid (Δ-8-THCA) by isomerization, or convert into (-)-Δ9-trans-tetrahydrocannabinol (THC) by decarboxylation [6]. Similarly, CBDA may convert into cannabinodiolic acid (CBNDA) through oxidation, into cannabielsoic acid (CBEA) via photochemical reactions, and into cannabidiol (CBD) through decarboxylation [6]. During the storage period, humidity, temperature, and exposure to light have been demonstrated in numerous studies to affect the bioactive compound composition, accelerating cannabinoid and terpene degradation reactions [1,6,18,19,20,21,22]. On the other hand, to the best of our knowledge, the influence of other variables such as the gas atmosphere during storage or vacuum storage has not been thoroughly explored yet. The incentive for examining the effects of modified low oxygen atmosphere and vacuum storage stems from the widely recognized phenomenon where oxygen presence during storage frequently accelerates the decay process [22,23]. This acceleration is notably observed in the oxidation of secondary metabolites, consequently leading to diminished storability. 

Hence, the current study aims to examine the effect of a combination of pre-harvest hexanoic acid priming and vacuum storage on the shelf life of cannabis inflorescences, under established optimal temperature and humidity conditions. Towards this end, we concentrated on the quantification of major cannabinoids and terpenes over a four-month storage period.

## 2. Results and Discussion

### 2.1. Effect of Pre-Harvest Hexanoic Acid Spray Application and Vacuum Storage on the Cannabinoid Content

The cannabinoid content within the high THCA chemovar was affected by the hexanoic acid treatment (Figure 1). On the day of harvest (t_0_), the concentrations of CBGA, CBDA, and total minor cannabinoids (cannabinoids with concentration below 1 DW%) were significantly higher by 16–22% in the inflorescences treated with hexanoic acid (Figure 1). However, at d6, following the drying and curing procedures, only the CBGA concentration was significantly elevated by 14% in the hexanoic acid treatment group in comparison to the reference group (Figure 1). The higher CBGA concentration at t_0_ and d6 in the hexanoic acid treatment group supports the hypothesis that pre-harvest hexanoic acid treatment on cannabis inflorescence can enhance the formation of major cannabinoids such as CBGA. This observation also provides an explanation for the higher cannabinoid concentrations observed in the hexanoic acid treatment group, as CBGA serves as the primary metabolite to form THCA, CBDA, and CBCA [6,7]. 

Furthermore, no statistically significant alterations were detected in the concentrations of the majority of the cannabinoids examined throughout the drying and curing processes, with the exception of THC concentration in the reference group, which increased by 47%, and CBG concentration in the hexanoic acid group, which decreased by 27% (Figure 2, Appendix A). In both cases, the absolute concentration changes were less than 0.08 DW% (Appendix A).

At the end of the storage period (d130), a noticeable difference in cannabinoid levels was found among the different groups studied (Figure 1). The highest level of the main cannabinoid, THCA, on d130 was found in the Hex8mM-VAC group, followed by REF-VAC, Hex8mM-air, and REF-air (Figure 1a). Hex8mM-VAC emerged as the only treatment group wherein the THCA concentration was significantly higher (by 38%) compared to the reference group stored in polyethylene plastic sealed packages (REF-air, Figure 1a). These findings imply that for the preservation of elevated THCA concentrations over an extended duration, the fine-tuning of both pre- and post-harvest processes is required. Additionally, the Hex8mM-VAC group exhibited the highest concentration of CBGA, CBDA, and CBCA, after four months of storage (22–45% higher than REF-air, Figure 1b–d). No significant differences in the concentration of the remaining minor cannabinoids among the different groups were observed (Figure 1e–h). Consequently, the highest total cannabinoid, total minor cannabinoid, and total THC content at d130 were observed in the Hex8mM-VAC group (Figure 1k,l). 

Following four months of storage (d130), the concentration of THCA, acting as the “reservoir” for THC, decreased in the various groups examined by 4–23%, with the minimal and statistically non-significant reduction being recorded in the Hex8mM-VAC group, in comparison to the conclusion of the drying and curing phase (d6, Figure 2a). On the other hand, the biggest decrease was observed in the REF-air group, highlighting the need for proper storage and pre-harvest treatment (Figure 2a). This latter observation aligns with the results documented in prior studies, where a 15–20% decrease in THCA concentration was noted post-storage under similar conditions [20,21], as well as with a recent study revealing 15–30% lower THCA concentrations in dried cannabis inflorescence as compared to the levels indicated on dispensed package labels [24]. Consequently, a revision of the prevailing industrial storage conditions is warranted. Surprisingly, the reduction in THCA levels does not result in a corresponding increase in THC or in any other hereby analyzed degradation products of THCA [6]. The sole THCA degradation product identified, CBNA, yielded a negligible concentration of approximately 0.02 DW% after four months of storage (Figure 1i). The absence of mass balance concerning THCA decline was similarly noted in other studies [19,20,21]. However, in stark contrast to previous findings [6,12,19,25], the decarboxylation of acidic cannabinoids to neutral cannabinoids was not observed in the current study during the four-month storage period. On the contrary, the concentrations of neutral cannabinoids, THC and CBG, significantly declined by 22–78% during storage (Figure 2g,h). Due to the concentration decrease in both THCA and THC during storage, the total THC content also decreased in the REF-air and REF-VAC groups (Figure 2l). Apart from CBCA and (-)-Δ9-trans-tetrahydrocannabiorcolic-C4 acid (THCA-C4), which sustained a stable concentration throughout storage in all groups studied, the remaining cannabinoids displayed a significant concentration decline during storage, with the most pronounced decrease noted in the REF-air group and the slightest decrease observed in the Hex8mM-VAC group (Figure 2). Taken altogether, the treatment group that yielded the highest cannabinoid concentration after four months of storage was the Hex8mM-VAC group.

Given that cannabis-based products are classified as medicinal products, they are subjected to strict medical grade regulations [26]. According to the EU guidelines for the shelf life estimation of drug substances, the shelf life of a medical product refers to the time period during which the product retains 95% of its major cannabinoids’ content determined at the beginning of the storage [17]. Consequently, solely the inflorescences from the Hex8mM-VAC treatment group, which exhibited a 96% relative THCA concentration at the end of the storage period, are considered unexpired, indicating that they remain suitable for the intended use for a minimum of four months (Figure 2a). Furthermore, the THCA concentration in the Hex8mM-VAC treatment group at the end of the storage exceeded the initial concentration of the reference groups at d6 by 6%. All other treatment groups exhibited a 77–90% relative THCA concentration at the end of the storage period and are considered expired (Figure 2a). These findings indicate that optimal pre-harvest conditions coupled with optimal post-harvest storage increased the product’s shelf life and consequently the medicinal efficacy of the final product. 

Recent studies investigating the influence of integrated parameters on cannabinoids’ stability during storage mainly focused on post-harvest parameters—e.g., the combination of temperature and light exposure [19,20,21,22,27]—while our study is the first to examine the specific integrated effect of both pre- and post-harvest approaches, which have not been considered so far. Moreover, the hexanoic acid spray results obtained in this study are in line with the impact of hexanoic acid on the secondary metabolite composition in other plant species studied such as tomato and citrus [13,15]. 

### 2.2. Effect of Pre-Harvest Hexanoic Acid Spray Application and Vacuum Storage on the Terpene Content

The influence of hexanoic acid treatment on the mono-terpene content within the high THCA chemovar was less pronounced than in the case of cannabinoids (Figure 3). At harvest day (t_0_), the concentrations of all mono-terpenes were similar and not statistically different in both reference and hexanoic acid treatment groups (Figure 3). Moreover, no differences in the concentrations of these mono-terpenes were observed between the reference and the hexanoic acid treatment groups after the drying and curing processes (d6, Figure 3), except for α-terpinenol which displayed a significantly higher concentration by 25% in the hexanoic acid treatment group than the reference group (Figure 3h). 

At the end of the drying and curing processes (d6), the concentration of d-limonene and β-myrcene decreased by 21–48% in all of the treatment groups as compared to t_0_, with the reference group displaying lower concentration values than the hexanoic acid treatment group (Figure 4a,d, Appendix A). No significant change in α-pinene concentrations was observed between t_0_ and the end of the drying and curing processes (d6, Figure 4b). On the other hand, at d6, the concentrations of (-)-β-pinene, linalool, fenchol, pinalol, and α-terpinenol decreased significantly by 13–34% only in the reference group (Figure 4c,e–h, Appendix A), indicating that the hexanoic acid pre-harvest treatment maintained steady-state levels of mono-terpenes during the entire drying and curing period. 

At the end of the storage period (d130), substantial differences in the mono-terpene content between the different groups were found (Figure 3). Except β-myrcene, the highest mono-terpene concentrations at d130 were observed in the Hex8mM-VAC group, followed by Hex8mM-air, REF-VAC, and REF-air groups (Figure 3). The concentrations of the various mono-terpenes in the Hex8mM-VAC group at d130 were higher by 20–82% as compared to the REF-air group (Figure 3). Moreover, both hexanoic acid pre-harvest treatment groups (Hex8mM-VAC and Hex8mM-air) exhibited significantly higher concentrations of the mono-terpenes d-limonene, (-)-β-pinene, linalool, and fenchol as compared to both reference groups stored under vacuum and air conditions (REF-VAC and REF-air, Figure 3a,c,e,f), indicating that hexanoic acid treatment maintained steady-state levels of the aforementioned aroma compounds during the storage period. The preservation of major aroma components is of the utmost importance, due to their contribution to the overall pharmacological activity of cannabis products as well as the appealing aroma of the inflorescence, specifically sought by both patients and recreational users. 

After four months of storage, the concentrations of α-pinene, (-)-β-pinene, and pinalol remained stable across all examined groups, with the exception of (-)-β-pinene in the REF-air group and pinalol in the Hex8mM-air group, where a statistically significant decrease of 13–14% was observed (Figure 4b,c,g). On the other hand, the concentrations of β-myrcene, fenchol, and α-terpineol significantly decreased in all of treatment groups by 19–51%, with the lowest reduction observed in the Hex8mM-VAC group (Figure 4d,f,h). The Hex8mM-VAC group was the only treatment group in which the concentration reduction of the major mono-terpene, d-limonene, was not statistically significant as compared to d6 (Figure 4a). Taken altogether, the integrated effect of the pre-harvest hexanoic acid treatment followed by vacuum storage under optimal temperature and humidity conditions yielded stable major cannabinoid and mono-terpene concentrations, thereby substantially increasing the products’ shelf life and therefore maintaining the quality and efficacy of the cannabis inflorescence. 

The hexanoic acid treatment affected the sesqui-terpene content differently than the mono-terpene content (Figure 5). At harvest day (t_0_), the concentrations of β-caryophyllene, α-humulene, (-)-α-bisabolol, γ-elemene, and α-selinene were significantly higher in the reference group as compared to the hexanoic acid treatment group (Figure 5a,b,d,e,i). Nevertheless, these differences were not maintained at the end of the drying and curing period (d6) due to a significant decrease of 13–21% in their concentrations, observed solely in the reference group (Figure 5a,b,d,e,i, Figure 6a,b,d,e,i). Moreover, at d6, β-eudesmene displayed a significant concentration decline of 21% as compared to t_0_ only in the reference group (Figure 6f, Appendix A). These results support the observation that the hexanoic acid pre-harvest treatment maintained steady-state levels of terpenes during the entire drying and curing period.

At the end of the storage period (d130), the concentration differences of several sesqui-terpenes between the Hex8mM-VAC group and the REF-air group mirrored those observed for both mono-terpenes and cannabinoids (Figure 5). The concentration of the sesqui-terpenes, β-caryophyllene, (-)-guaiol, γ-eudesmol, β-eudesmol, and α-selinene were significantly larger only in the Hex8mM-VAC group as compared to the REF-air group by 19–43% (Figure 5a,c,g–i). On the other hand, no significant differences were observed in the concentration of α-humulene, (-)-α-bisabolol, γ-elemene, and β-eudesmene after four months of storage between the different groups (Figure 5). 

At d130, the concentration of the sesqui-terpenes, α-humulene, (-)-guaiol, γ-elemene, β-eudesmene, and γ-eudesmol remained stable and not statistically significant as compared to d6 in all treatment groups studied (Figure 6). On the other hand, β-caryophyllene and β-eudesmol displayed a significant concentration decline (by 15–22%) only under non-vacuum storage conditions, (-)-α-bisabolol displayed a significant concentration decline (by 19–22%) in all groups examined except REF-air, while α-selinene exhibited a significant concentration decrease in all treatment groups (by 23–43%, Figure 6). Taken altogether, these results suggest that sesqui-terpenes are less susceptible to degradation than mono-terpenes during long storage times. Moreover, these results suggest that sesqui-terpenes are less influenced by the integrated effect of pre-harvest hexanoic acid treatment and vacuum storage.

Significant differences in total terpene content between the treatment groups were found only after four months of storage. The highest mono-terpene content was observed in the Hex8mM-VAC group, followed by the Hex8mM-air group, the REF-VAC group, and the REF-air group (Figure 7a). This indicates that hexanoic acid treatment had a greater impact on the mono-terpene content preservation than storage under vacuum without pre-harvest spraying. During drying and curing, the mono-terpene content in all groups examined, except the Hex8mM-VAC group, resulted in an 18–26% significant decrease (Figure 8a). On the other hand, no significant differences in sesqui-terpene content were observed between the different groups under all time points investigated, possibly since they are less susceptible to degradation and are less volatile than the mono-terpenes (Figure 7b) [28]. The highest total terpene content after four months of storage was observed in the Hex8mM-VAC group, which was the only treatment group that was significantly different from the REF-air group (Figure 7c).

The literature reports about the impact of various post-harvest conditions on terpenes’ preservation in cannabis inflorescence are conflicting [19,22]. Grafström et al. found that the concentration of numerous sesqui-terpenes remained stable during four months of airtight storage conditions, in agreement with our study outcome [22]. On the other hand, Milay et al. found that the mono- and sesqui-terpene content significantly decreased after four months of storage regardless of the storage temperature [19]. 

The application of hexanoic acid to citrus trees resulted in an enhanced content of aromatic volatile compounds [13]. However, in the current study, this effect was only discerned in the cannabis inflorescence at the conclusion of the storage period and not during the drying and curing stages. Additionally, it was observed that the reference group underwent a significant reduction in 11 distinct aroma compounds during drying and curing, whereas the hexanoic acid group did not exhibit such a decrease. These findings imply that hexanoic acid treatment in cannabis may aid in the preservation of volatile compounds during the initial post-harvest stages and throughout storage, rather than increasing their production as observed in citrus [13]. 

The combined application of pre-harvest hexanoic acid spraying and post-harvest vacuum storage has shown to either match or enhance cannabis inflorescence shelf life in terms of major cannabinoid retention [19,20,21]. Trofin et al. observed a 17% reduction in THCA concentration after five months of storage at 4 °C, whereas Reason et al. noted a 10% reduction in THCA concentration after two months of storage at 6 °C [20,21]. Compared to these findings, our Hex8mM-VAC treatment exhibited a reduced THCA degradation, with only a 4% decrease in relative concentration after four months.

Milay et al. documented a 15% drop in THCA relative concentration over 12 months of storage at 4 °C, translating to a 5% decrease over a four-month period, slightly more than our 4% decrease following Hex8mM-VAC treatment [19]. Furthermore, a significant reduction in terpene levels during four months of storage at 4 °C has been reported in Milay et al. [19]. Milay et al. found a 39–64% reduction in the concentrations of α-pinene, (-)-β-pinene, β-myrcene, linalool, fenchol, d-limonene, α-terpinenol, β-caryophyllene, α-humulene, and (-)-α-bisabolol after four months of storage at 4 °C [19]. In contrast, our Hex8mM-VAC method resulted in a less pronounced decline in terpenes, with decreases of 20–23% for fenchol, α-terpinenol, and (-)-α-bisabolol; a 37% decrease for β-myrcene; while the levels of α-pinene, (-)-β-pinene, linalool, d-limonene, β-caryophyllene, and α-humulene remained constant throughout the storage period. Thus, the integrated effect of hexanoic acid spraying and vacuum storage appears to surpass the performance of storage conditions reported in the literature.

## 3. Materials and Methods

### 3.1. Chemicals

Acetonitrile, anhydrous ammonium formate, ethanol, formic acid, and hexanoic acid (>99%) were obtained from Sigma-Aldrich (HPLC grade, Saint Louis, MO, USA). Ultra-pure water was provided by the Milli-Q Plus system (Millipore Corp., Billerica, MA, USA). Cannabinoid analytical standards were purchased from RESTEK (RESTEK, Bellefonte, PA, USA): cannabidivarinic acid (CBDVA), cannabigerovarinic acid (CBGVA), cannabidiolic acid (CBDA), cannabigerolic acid (CBGA), cannabigerol (CBG), cannabidiol (CBD), (-)-Δ9-trans-tetrahydrocannabivarin (THCV), (-)-Δ9-trans-tetrahydrocannabivarinic acid (THCVA), cannabinol (CBN), cannabinolic acid (CBNA), cannabichromevarinic acid (CBCVA), (-)-Δ9-trans-tetrahydrocannabinol (Δ-9-THC), (-)-Δ8-trans-tetrahydrocannabinol (Δ-8-THC), cannabicyclol (CBL), cannabicyclolic acid (CBLA), cannabichromene (CBC), (-)-Δ9-trans-tetrahydrocannabinolic acid (THCA), and cannabichromenic acid (CBCA). Each of those standards was obtained at a stock concentration of 1000 µg/mL except CBLA which was obtained at a stock concentration of 500 µg/mL. Terpene standard mix at a stock concentration of 2500 µg/mL from each terpene containing the following terpenes—α-pinene, camphene, (-)-β-pinene, β-myrcene, δ-3-carene, α-terpinene, p-cymene, d-limonene, ocimene, γ-terpinene, terpinolene, linalool, (-)-isopulegol, geraniol, β-caryophyllene, α-humulene, nerolidol, (-)-guaiol, and (-)-α-bisabolol—was obtained from RESTEK (RESTEK, Bellefonte, PA, USA).

### 3.2. Plant Material

Freshly harvested medicinal *Cannabis sativa* L. inflorescences from commercially available high THCA chemovar, “505”, were provided by the Bar-Lev farm in December 2022 on the same day of harvest (Bar-Lev Agricultural Crops, Kfar Hess, Israel, 32°15′21.2″ N 34°57′01.0″ E). The inflorescences were analyzed for their cannabinoid and terpene content at the Agricultural Research Organization, the Department of Food Science. 

The cannabis plants were cultivated at the Bar-Lev farm greenhouse over a period of 86 days. Initially, the plants were subjected to a vegetative growth phase for 30 days, during which they received 18 h of light and 6 h of darkness daily. This phase was succeeded by a 56-day flowering period, characterized by an equal photoperiod of 12 h of light and 12 h of darkness. Throughout the entire growth cycle, the plants were maintained under ambient temperature conditions, with relative humidity levels consistently held between 40% and 60%. The plants were established in a detached planting bed, utilizing a growth medium composed of a 70:30 ratio of peat and tuff. 

### 3.3. Pre-Harvest Hexanoic Acid Spray Application

The stock solution of hexanoic acid at a concentration of 0.93 g/mL was diluted with appropriate volumes of water to produce an 8 mM working solution. Hexanoic acid (“Hex8mM”) was applied on 30 apical cannabis inflorescences collected from 10 plants by manual spray application three and seven days before harvest. Each inflorescence was sprayed three times with 20 mL of 8 mM hexanoic acid (Hex8mM). The spray application schedule was determined to be optimal in preliminary experiments when compared to a single spray application either three or seven days prior to harvest. Furthermore, based on past research, the initiation of plant secondary metabolite synthesis pathways typically requires about three to four days to achieve peak levels, which explains the aforementioned application frequency [29,30]. The reference group (“REF”) consisting of 30 apical cannabis inflorescences from 10 plants was sprayed only with water under the same conditions as for the treatment group. It was verified in preliminary experiments that there was no statistical difference in the inflorescence cannabinoid and terpene concentrations due to water spray as compared to “no spraying” control inflorescence. 

The optimal concentration of hexanoic acid (8 mM) was determined in a set of preliminary experiments, in which four different concentrations (1, 4, 8, and 10 mM) were sprayed on cannabis inflorescence and the concentration of THCA, CBDA, and CBGA was determined after drying and curing processes (d6, Appendix A). 

### 3.4. Drying and Curing Processes

To determine the endpoint of the drying process, we quantified the total water content loss during drying. Initially, the absolute water loss was established by drying samples at 105 °C in a dry oven (DFO-150, MRC, Harlow, UK). This value served as a benchmark to identify the corresponding point under our experimental drying conditions when the same degree of water loss occurred. To ensure the validity of our approach, preliminary tests were conducted, confirming no significant statistical difference (*p* > 0.05) between water content measurements obtained by these two methods.

Controlled atmospheric conditions were applied using controlled atmosphere drying chambers (60 × 65 × 105 cm^3^, SCS, Paddock Wood, UK) for the initial drying and curing processes of the fresh cannabis inflorescences. To attain a relative humidity of less than 10% in the drying chambers, we used O_2_, N_2_, and CO_2_ gases with humidity levels below 0.1%. The measured relative humidity was on average 8.5%. Additionally, within the controlled atmosphere drying chambers, we inserted 500 g of dried silica gel pearls (Drying pearls orange, Merck, Germany) to absorb the moisture during the drying process. The temperature in the drying chambers was set to 15 °C, within the commonly used drying temperature of 15–21 °C [1]. The controlled atmosphere drying chamber’s automated system (ICA6000, ICAstorage, Paddock Wood, UK) restored the desired drying and curing conditions within the chambers every 30 min. After six days (d6), the cannabis inflorescences were completely dry, since no additional inflorescence weight loss was observed (the variation in weight fell within the margin of error for the water content, which is 10% ± 1% for the dried cannabis inflorescence) [31]. During the drying procedure, all of the cannabis inflorescences were placed in breathable trays within the controlled atmosphere chambers. In order to determine the water content of the cannabis inflorescence samples, three plates containing weighted cannabis inflorescences were placed in the drying chambers and the weight was recorded before, during, and after drying for both REF and Hex8mM groups. Each plate contained three cannabis inflorescences weighting 5–7 g. For calculating the cannabinoid and terpene content, the cannabis fresh inflorescence weight at the harvest day (t_0_) was normalized to dry weight using the average weight loss. The average weight loss during drying and curing (d6) was 80.5% ± 1.5%. 

### 3.5. Storage

Cannabis inflorescences were stored for four months (d130) in the dark. The inflorescences were stored either in a polyethylene plastic sealed package (“air”, CD-200 impulse heat sealer, Carerite, Woodland Hills, CA, USA) or under vacuum (“VAC”, Chemical Duty Vacuum Pressure Pump, MilliporeSigma, Burlington, MA, USA). The storage temperature was set to 4 °C, which was found to be superior in terms of cannabinoid and terpene content preservation for a prolonged storage time as compared to the other temperatures examined previously [19,20]. 

During storage, four different groups were examined, namely, the reference group that was stored in a polyethylene plastic sealed package (“REF-air”), the reference group that was stored under vacuum (“REF-VAC”), the hexanoic acid group that was stored in a polyethylene plastic sealed package (“Hex8mM-air”), and the hexanoic acid group that was stored under vacuum (“Hex8mM-VAC”).

### 3.6. Sample Preparation

At each time point (i.e., t_0_, d6, and d130), fresh (t_0_) or dried (d6 and d130) cannabis inflorescences from the high THCA chemovar were ground homogenously with a mortar and pestle in the presence of liquid nitrogen, providing 5 replicates from each group. The homogenously ground cannabis samples (500 ± 0.5 mg for fresh inflorescences and 100 ± 0.1 mg for dried inflorescences) were extracted with 4 mL of ethanol in 15 mL Falcon tubes and shaken (Digital Orbital Shaker, MRC, Holon, Israel) in the dark for 15 min at 500 rpm. An amount of 1 mL of the extract was transferred to an Eppendorf tube and centrifuged for 4 min at 12,000 rpm. For the determination of cannabinoid levels, a dilution of 1:5 of the supernatant with ethanol was carried out, and 1 mL aliquot was transferred to an HPLC vial and subjected to high-pressure liquid-chromatography-photodiode array (HPLC-PDA) analysis. For the determination of terpene levels, 0.25 mL of the supernatant was inserted into a GC vial and analyzed via gas chromatography–mass spectroscopy (GC/MS).

### 3.7. Quantification of Cannabinoids by HPLC-PDA and Terpenes by GC/MS

The ethanolic cannabis extracts were analyzed as described in Birenboim et al., utilizing HPLC-PDA (Acquity Arc FTN-R; Model PDA-2998, Waters Corp., Milford, MA, USA) equipped with a Kinetex^®^ 1.7 μm XB-C18 100A LC column (150 × 2.1 mm i.d. and 1.7 μm particle size; Phenomenex, Torrance, CA, USA) for the cannabinoids’ analysis [32]. Cannabinoids were quantified by comparing the integrated peak area with the corresponding cannabinoid calibration curve ranging from 1 to 1000 µg/mL [32]. To ensure the precise identification of each cannabinoid present in the ethanolic cannabis extract, we conducted comparisons of both the peak’s retention time and the UV spectra against those of the corresponding standards [32]. To express concentrations in terms of dry weight% (DW%), we transformed the measured concentrations from µg/mL based on the extract’s dilution factor and the precise weight recorded for each sample, employing Equation (1) for the conversion. Additionally, we have made a distinction between the following cannabinoids: minor cannabinoids, which are present at concentrations <1 DW%, and major cannabinoids, found at concentrations of ≥1 DW%. The terpene analysis was carried out by GC/MS (Agilent, Santa Clara, CA, USA) as recently reported by Birenboim et al. utilizing a DB-5 capillary column (5% phenyl, 95% dimethylpolysiloxane, 30 m × 0.250 mm, 0.25 m; Agilent, Santa Clara, CA, USA) for analyte separation [32]. Terpenes were quantified by comparing the integrated peak area with the corresponding terpene calibration curve ranging from 0.5 to 250 µg/mL [32]. Representative HPLC-PDA and GC/MS chromatograms together with the identification of the main peaks is provided in the Appendix A. The methods’ analytical validation parameters (i.e., R^2^, limit of detection, limit of quantification, repeatability, and accuracy) were recently published in Birenboim et al. [32].
(1)$$DW\%=100\%*\frac{concentration\left[\frac{µg}{mL}\right]*dilution\;factor[mL]}{sample\;weight\left[mg\right]*\frac{1000µg}{1mg}}$$

### 3.8. Statistical Analysis and Cannabinoid/Terpene Content Calculations 

For each compound analyzed, the multiple comparison tool of two-way ANOVA followed by Tukey’s post hoc test was used to determine the differences in cannabinoid and terpene concentrations between the different groups and the reference group (REF-air) at each time point, at α = 0.05 using GraphPad PRISM 10 (San Diego, CA, USA). Moreover, within each group, the multiple comparison tool of two-way ANOVA followed by Tukey’s post hoc test was used to determine the differences in cannabinoid and terpene concentrations between each time point, at α = 0.05 using GraphPad PRISM 10 (San Diego, CA, USA).

Total THC content was calculated according to Equation (2) [33].
(2)$$total\;THC\left[DW\%\right]=\left(THCA\;DW\%\right)\;\times \;0.877+THC\;DW\%$$

## 4. Conclusions

This study explored the impact of pre-harvest hexanoic acid spray application and post-harvest vacuum storage on the cannabinoid and terpene content of cannabis inflorescences over four months of storage. Hexanoic acid spray application yielded higher concentrations of certain cannabinoids and terpenes, CBGA, CBDA, and α-terpineol at harvest day as well as after drying and curing stages. Hexanoic acid treatment also contributed to terpene content preservation during drying and curing processes. Integrated treatment composed of pre-harvest hexanoic acid spray application and post-harvest storage under vacuum yielded the highest concentration of cannabinoids and mono-terpenes after four months of storage resulting in the longest shelf life as compared to all other groups examined. Although the concentration of most cannabinoids studied declined at the end of the storage period, a corresponding increase in the degradation products was not observed. Moreover, decarboxylation of acidic cannabinoids to neutral cannabinoids was not observed during storage. Pre-harvest hexanoic acid treatment was found to preserve total terpene content. In addition, hexanoic acid spray application displayed a more pronounced impact on mono-terpene preservation than storage under vacuum without hexanoic acid treatment. On the other hand, sesqui-terpenes were found to be less susceptible to degradation than mono-terpenes during four months of storage and less influenced by the integrated effect of hexanoic acid treatment and vacuum storage. Our results underscore the importance of combining pre- and post-harvest methodologies to extend the shelf life of cannabis inflorescence.

## Figures and Tables

**Figure 1 plants-13-00992-f001:**
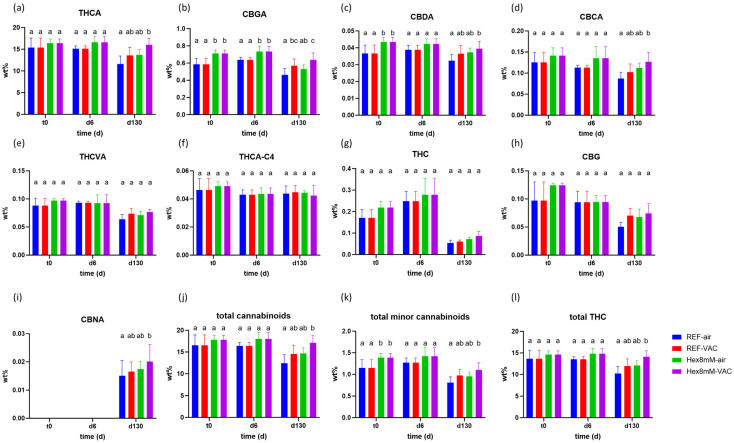
Mean cannabinoid concentrations (in DW%, y-axis, n = 5) in cannabis inflorescence under four different pre- and post-harvest conditions: REF-air (reference group that was stored in polyethylene plastic sealed package), REF-VAC (reference group that was stored under vacuum), Hex8mM-air (hexanoic acid group that was stored in polyethylene plastic sealed package), and Hex8mM-VAC (hexanoic acid group that was stored under vacuum). (**a**) THCA, (**b**) CBGA, (**c**) CBDA, (**d**) CBCA, (**e**) THCVA, (**f**) THCA-C4, (**g**) THC, (**h**) CBG, (**i**) CBNA, (**j**) total cannabinoids, (**k**) total minor cannabinoids and (**l**) total THC. Cannabinoid concentrations at t_0_ were normalized to DW% using mean weight loss. Statistical significance between the different groups and the reference group (REF-air) for each compound at each time point was calculated using two-way ANOVA followed by Tukey’s post hoc test at α = 0.05. Letters above each bar are used to present the statistical test results at each time point.

**Figure 2 plants-13-00992-f002:**
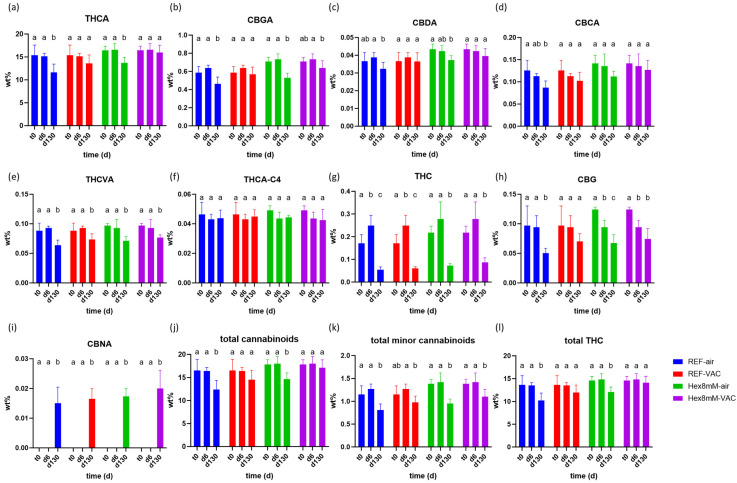
Mean cannabinoid concentrations (in DW%, y-axis, n = 5) in cannabis inflorescence under four different pre- and post-harvest conditions: REF-air (reference group that was stored in polyethylene plastic sealed package), REF-VAC (reference group that was stored under vacuum), Hex8mM-air (hexanoic acid group that was stored in polyethylene plastic sealed package), and Hex8mM-VAC (hexanoic acid group that was stored under vacuum). (**a**) THCA, (**b**) CBGA, (**c**) CBDA, (**d**) CBCA, (**e**) THCVA, (**f**) THCA-C4, (**g**) THC, (**h**) CBG, (**i**) CBNA, (**j**) total cannabinoids, (**k**) total minor cannabinoids and (**l**) total THC. Cannabinoid concentrations at t_0_ were normalized to DW% using mean weight loss. Statistical significance between the different time points within each treatment group for each compound was calculated using two-way ANOVA followed by Tukey’s post hoc test at α = 0.05. Letters above each bar are used to present the statistical test results within each treatment group.

**Figure 3 plants-13-00992-f003:**
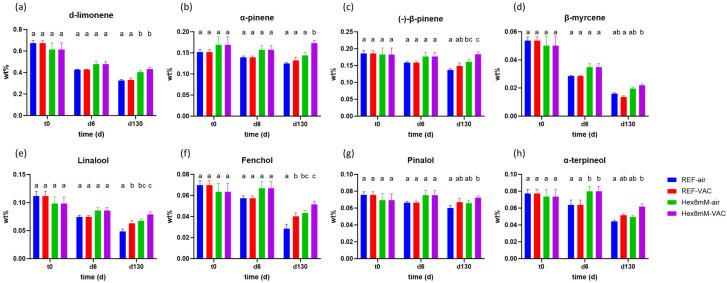
Mean mono-terpene concentrations (in DW%, y-axis, n = 5) in cannabis inflorescence under four different pre- and post-harvest conditions: REF-air (reference group that was stored in polyethylene plastic sealed package), REF-VAC (reference group that was stored under vacuum), Hex8mM-air (hexanoic acid group that was stored in polyethylene plastic sealed package), and Hex8mM-VAC (hexanoic acid group that was stored under vacuum). (**a**) d-limonene, (**b**) α-pinene, (**c**) (-)-β-pinene, (**d**) β-myrcene, (**e**) linalool, (**f**) fenchol, (**g**) pinalol, and (**h**) α-terpinenol. Mono-terpene concentrations at t_0_ were normalized to DW% using mean weight loss. Statistical significance between the different groups and the reference group (REF-air) for each compound at each time point was calculated using two-way ANOVA followed by Tukey’s post hoc test at α = 0.05. Letters above each bar are used to present the statistical test results at each time point.

**Figure 4 plants-13-00992-f004:**
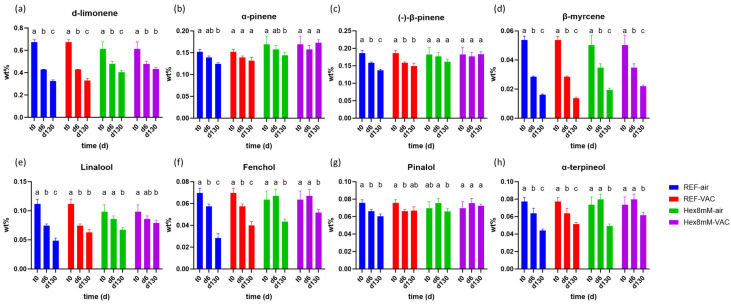
Mean mono-terpene concentrations (in DW%, y-axis, n = 5) in cannabis inflorescence under four different pre- and post-harvest conditions: REF-air (reference group that was stored in polyethylene plastic sealed package), REF-VAC (reference group that was stored under vacuum), Hex8mM-air (hexanoic acid group that was stored in polyethylene plastic sealed package), and Hex8mM-VAC (hexanoic acid group that was stored under vacuum). (**a**) d-limonene, (**b**) α-pinene, (**c**) (-)-β-pinene, (**d**) β-myrcene, (**e**) linalool, (**f**) fenchol, (**g**) pinalol, and (**h**) α-terpinenol. Mono-terpene concentrations at t_0_ were normalized to DW% using mean weight loss. Statistical significance between the different time points within each treatment group for each compound was calculated using two-way ANOVA followed by Tukey’s post hoc test at α = 0.05. Letters above each bar are used to present the statistical test results within each treatment group.

**Figure 5 plants-13-00992-f005:**
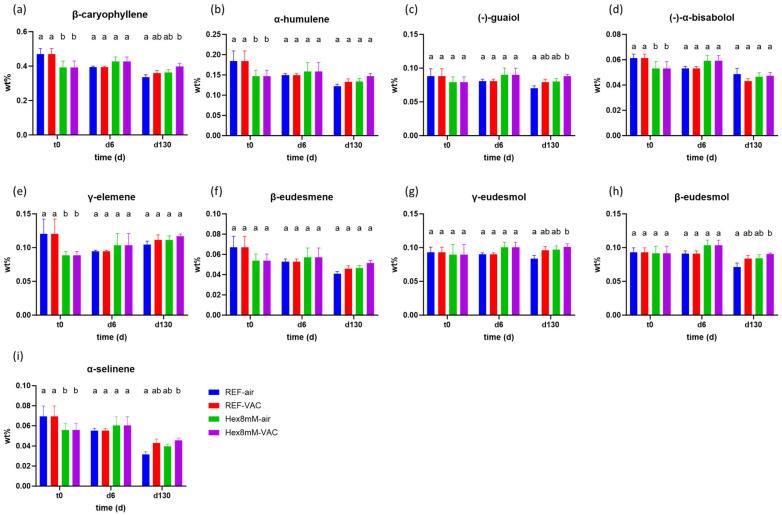
Mean sesqui-terpene concentrations (in DW%, y-axis, n = 5) in cannabis inflorescence under four different pre- and post-harvest conditions: REF-air (reference group that was stored in polyethylene plastic sealed package), REF-VAC (reference group that was stored under vacuum), Hex8mM-air (hexanoic acid group that was stored in polyethylene plastic sealed package), and Hex8mM-VAC (hexanoic acid group that was stored under vacuum). (**a**) β-caryophyllene, (**b**) α-humulene, (**c**) (-)-guaiol, (**d**) (-)-α-bisabolol, (**e**) γ-elemene, (**f**) β-eudesmene, (**g**) γ-eudesmol, (**h**) β-eudesmol and (**i**) α-selinene. Sesqui-terpene concentrations at t_0_ were normalized to DW% using mean weight loss. Statistical significance between the different groups and the reference group (REF-air) for each compound at each time point was calculated using two-way ANOVA followed by Tukey’s post hoc test at α = 0.05. Letters above each bar are used to present the statistical test results at each time point.

**Figure 6 plants-13-00992-f006:**
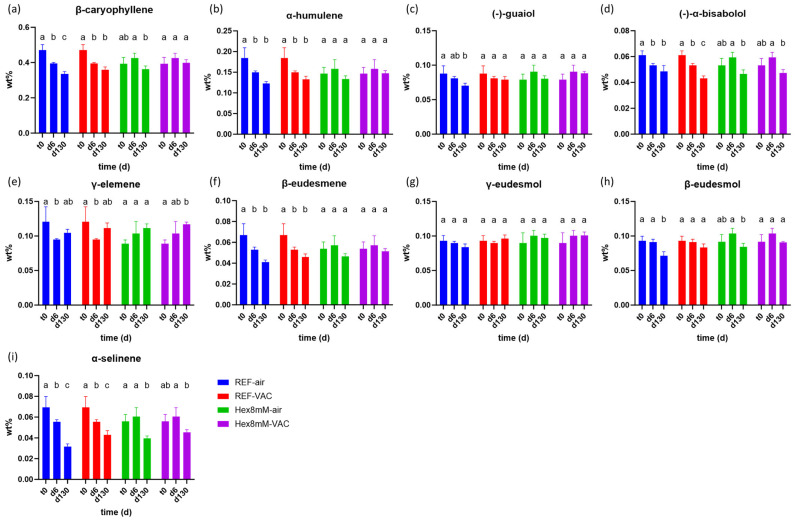
Mean sesqui-terpene concentrations (in DW%, y-axis, n = 5) in cannabis inflorescence under four different pre- and post-harvest conditions: REF-air (reference group that was stored in polyethylene plastic sealed package), REF-VAC (reference group that was stored under vacuum), Hex8mM-air (hexanoic acid group that was stored in polyethylene plastic sealed package), and Hex8mM-VAC (hexanoic acid group that was stored under vacuum). (**a**) β-caryophyllene, (**b**) α-humulene, (**c**) (-)-guaiol, (**d**) (-)-α-bisabolol, (**e**) γ-elemene, (**f**) β-eudesmene, (**g**) γ-eudesmol, (**h**) β-eudesmol and (**i**) α-selinene. Sesqui-terpene concentrations at t_0_ were normalized to DW% using mean weight loss. Statistical significance between the different time points within each treatment group for each compound was calculated using two-way ANOVA followed by Tukey’s post hoc test at α = 0.05. Letters above each bar are used to present the statistical test results within each treatment group.

**Figure 7 plants-13-00992-f007:**
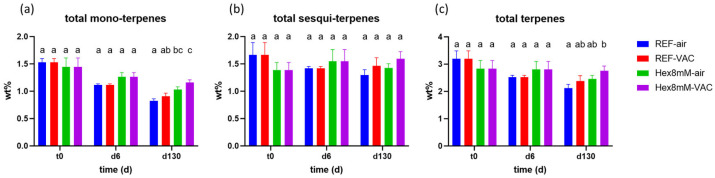
Mean total terpene content (in DW%, y-axis, n = 5) in cannabis inflorescence under four different pre- and post-harvest conditions: REF-air (reference group that was stored in polyethylene plastic sealed package), REF-VAC (reference group that was stored under vacuum), Hex8mM-air (hexanoic acid group that was stored in polyethylene plastic sealed package), and Hex8mM-VAC (hexanoic acid group that was stored under vacuum). (**a**) total mono-terpenes, (**b**) total sesqui-terpenes, and (**c**) total terpenes. Total terpene content at t_0_ was normalized to DW% using mean weight loss. Statistical significance between the different groups and the reference group (REF-air) for each compound at each time point was calculated using two-way ANOVA followed by Tukey’s post hoc test at α = 0.05. Letters above each bar are used to present the statistical test results at each time point.

**Figure 8 plants-13-00992-f008:**
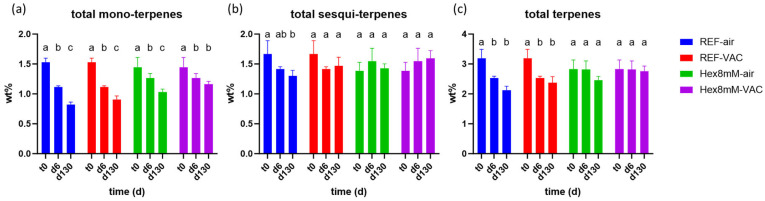
Mean total terpene content (in DW%, y-axis, n = 5) in cannabis inflorescence under four different pre- and post-harvest conditions: REF-air (reference group that was stored in polyethylene plastic sealed package), REF-VAC (reference group that was stored under vacuum), Hex8mM-air (hexanoic acid group that was stored in polyethylene plastic sealed package), and Hex8mM-VAC (hexanoic acid group that was stored under vacuum). (**a**) total mono-terpenes, (**b**) total sesqui-terpenes, and (**c**) total terpenes. Total terpene content at t_0_ was normalized to DW% using mean weight loss. Statistical significance between the different time points within each treatment group for each compound was calculated using two-way ANOVA followed by Tukey’s post hoc test at α = 0.05. Letters above each bar are used to present the statistical test results within each treatment group.

## Data Availability

Data will be made available on request.

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
