# Peer review of "Improved Long-Term Preservation of Cannabis Inflorescence by Utilizing Integrated Pre-Harvest Hexanoic Acid Treatment and Optimal Post-Harvest Storage Conditions"

_plants, 2024, doi:10.3390/plants13070992_

Round 1

Reviewer 1 Report

Comments and Suggestions for Authors

In the manuscript investigations on improved long term Preservation of Cannabis inflorescence by application of pre-harvest treatment by hexanoic acid and optimal post-harvest storage conditions has been described. The development of new procedures of obtaining and storing plant materials containing biologically active compounds is an important aspects of scientific research. The obtained results may contribute to the development of optimal conditions for obtaining and storing plant material of Cannabis sativa used in medicine.

However, the following corrections or suggestions should be addressed before publication.

The manuscript lacks information on how to identify peaks of compounds determined by the HPLC-PDA. It should be clarified whether analytes were identified only on the retention times or also on the comparison of the UV-Vis spectra of standards and analytes in samples.

Exemplary HPLC-PDA and GC-MS chromatograms and MS spectra should be also added to the manuscript or the supplementary materials.

The advantages and novelty of the described investigations should be additionally emphasized especially compared with similar previously described.

Author Response

Comments to the reviewers

We would like to thank the editor and external reviewers for their thoughtful and detailed comments on our paper. We have edited the manuscript to address all of their concerns. We believe that the manuscript is now suitable for publication.

Reviewer 1

1) The manuscript lacks information on how to identify peaks of compounds determined by the HPLC-PDA. It should be clarified whether analytes were identified only on the retention times or also on the comparison of the UV-Vis spectra of standards and analytes in samples.

We appreciate the reviewer's observation. The identification of cannabinoids within the extracts was achieved by comparing the retention time of each peak to that of the corresponding standard, as well as by comparing the UV spectra of the peak to the UV spectra of the corresponding standard. To clarify, we have included this information in the manuscript on page 14, lines 503-506: "To ensure the precise identification of each cannabinoid present in the ethanolic cannabis extract, we conducted comparisons of both the peak's retention time and the UV spectra against those of the corresponding standards."

2) Exemplary HPLC-PDA and GC-MS chromatograms and MS spectra should be also added to the manuscript or the supplementary materials.

As requested, we have included representative HPLC-PDA and GC-MS chromatograms to the supplementary section (Figures S1 and S2).

3) The advantages and novelty of the described investigations should be additionally emphasized especially compared with similar previously described.

Upon request, we have incorporated a comparative analysis between our optimal storage outcomes and those documented in the literature (page 11, lines 366-385). "The combined application of pre-harvest hexanoic acid spraying and post-harvest vacuum storage has shown to either match or enhance cannabis inflorescence shelf life in terms of major cannabinoid retention. Trofin et al. observed a 17% reduction in THCA concentration after five months of storage at 4°C, whereas Reason et al. noted a 10% reduction in THCA concentration after two months of storage at 6°C. Compared to these findings, our Hex8mM-VAC treatment exhibited a reduced THCA degradation, with only a 4% decrease in relative concentration after four months. Milay et al. documented a 15% drop in THCA relative concentration over 12 months of storage at 4°C, translating to a 5% decrease over a four-month period, slightly more than our 4% decrease following Hex8mM-VAC treatment. Despite the similar rates of THCA degradation, a significant reduction in terpene levels during four months of storage at 4°C has been reported in Milay et al.. Milay et al. found a 39-64% reduction in the concentrations of α-pinene, (-)-β-pinene, β-myrcene, linalool, fenchol, d-limonene, α-terpinenol, β-caryophyllene, α-humulene, and (-)-α-bisabolol after four months of storage at 4°C. In contrast, our Hex8mM-VAC method resulted in a less pronounced decline in terpenes, with decreases of 20-23% for fenchol, α-terpinenol, and (-)-α-bisabolol, a 37% decrease for β-myrcene, while the levels of α-pinene, (-)-β-pinene, linalool, d-limonene, β-caryophyllene, and α-humulene remained constant throughout the storage period. Thus, the integrated effect of hexanoic acid spraying and vacuum storage appears to surpass the performance of storage conditions reported in the literature."

Reviewer 2 Report

Comments and Suggestions for Authors

A fluent explanation of the treatment and aging process of
inflorescence. Supported by significant analytical work quantifying
various chemical compounds, none of which are given a unit of measure to
express concentration. The statistical treatment of the data used basic
tools such as ANOVA, although the authors refer to a two-way ANOVA,
either incorrectly indicating the chosen algorithm, or because they are
not sufficiently competent and therefore confuse basic statistical tools

Author Response

Comments to the reviewers

We would like to thank the editor and external reviewers for their thoughtful and detailed comments on our paper. We have edited the manuscript to address all of their concerns. We believe that the manuscript is now suitable for publication.

Reviewer 2

1) A fluent explanation of the treatment and aging process of inflorescence. Supported by significant analytical work quantifying various chemical compounds, none of which are given a unit of measure to express concentration.

Following the request, we have provided additional information on the aging process of cannabis during post-harvest storage on page 2, lines 75-82. "Based on existing literature, it is understood that both THCA and CBDA undergo various degradation pathways during storage. Specifically, THCA may degrade into cannabinolic acid (CBNA) or cannabitriolic acid (CBTA) through oxidation, transform into (-)-Δ8-trans-tetrahydrocannabinolic acid (Δ-8-THCA) by isomerization, or convert into (-)-Δ9-trans-tetrahydrocannabinol (THC) by decarboxylation. Similarly, CBDA may convert into cannabinodiolic acid (CBNDA) through oxidation, into cannabielsoic acid (CBEA) via photochemical reactions, and into cannabidiol (CBD) through decarboxylation."

Additionally, we clarified the units used to report concentration as dry weight percentage (DW%) on page 14, lines 506-509. "To express concentrations in terms of dry weight% (DW%), we transformed the measured concentrations from µg/mL based on the extract's dilution factor and the precise weight recorded for each sample, employing Equation 1 for the conversion."

2) The statistical treatment of the data used basic tools such as ANOVA, although the authors refer to a two-way ANOVA, either incorrectly indicating the chosen algorithm, or because they are not sufficiently competent and therefore confuse basic statistical tools

We are grateful for the reviewer's input. Additional clarification is indeed warranted. Our dataset is structured such that, for each compound analyzed, each column corresponds to a different treatment group, and each row corresponds to a different time point. As a result, we did not employ the two-way ANOVA to assess interactions between hexanoic acid treatment and vacuum storage. Instead, we used the multiple comparison feature within the two-way ANOVA framework to identify significant variations in cannabinoid and terpene concentrations across each treatment over different time points, and between treatments and reference group at each time point. Given our data structure, Tukey's posthoc test was applied, being the most suitable for our analysis. Since at each time point the samples belonging to the same treatment group were different, repeated ANOVA was not suitable test for us. This clarification has been incorporated into the manuscript on page 14, lines 526-533: " For each compound analyzed, the multiple comparison tool of two-way ANOVA followed by Tukey's posthoc test was used to determine the differences in cannabinoid and terpene concentrations between the different groups and the reference group (REF-air) at each time point, at α = 0.05 using GraphPad PRISM 10 (San Diego, CA, USA). Moreover, within each group, the multiple comparison tool of two-way ANOVA followed by Tukey's posthoc test was used to determine the differences in cannabinoid and terpene concentrations between each time point, at α = 0.05 using GraphPad PRISM 10 (San Diego, CA, USA)."

3) page 5 line 159: equation 1 can still be used to estimate the concentration of THC

THC concentration was determined using HPLC-PDA, with its variation across different treatments and time points illustrated in Figures 1g and 2g. As discussed on page 5, lines 162-172, it was observed that THC concentration declines over the storage period without any noticeable decarboxylation from THCA to THC. Additionally, we computed the total THC content using Equation 1, which accounts for both THCA and THC levels, and presented these findings in Figures 1 and 2l. It was noted that the total THC content similarly diminished during storage, a point we have elucidated further in the manuscript on page 5, lines 172-174.

On behalf of all co-authors, we thank the reviewers for their comments and valuable suggestions.

Jakob Shimshoni, PhD
Department of Food Science
Institute for Postharvest and Food Sciences
Agricultural Research Organization
Rishon LeZiyyon 
7528809
Israel

Round 2

Reviewer 2 Report

Comments and Suggestions for Authors

Actually I still belive that ANOVA was used instead of two way ANOVA.

Several bibliogrraphic links are not working, almost that was for my system.

Author Response

27/03/24

Comments to the reviewer

We have edited the manuscript to address all of their concerns. We believe that the manuscript is now suitable for publication.

Review 2

  1. "Actually I still belive that ANOVA was used instead of two way ANOVA".

We wish to assure the reviewer that indeed two-way ANOVA has been done. We have used for that purpose GraphPad Prism 10. For the reviewer's information, we have used the exactly the two way ANAOV described in the following website of GraphPad: https://www.graphpad.com/guides/prism/latest/statistics/stat_multiple_comparisons_tab_2way.htm

  1. "Several bibliographic links are not working, almost that was for my system"

We thank the reviewer for this note. We corrected all reference links as requested .

On behalf of all co-authors, we thank the reviewers for their comments and valuable suggestions.

Jakob Shimshoni, PhD
Department of Food Science
Institute for Postharvest and Food Sciences
Agricultural Research Organization
Rishon LeZiyyon 
7528809
Israel
